# A Real-Time Rescheduling Algorithm for Multi-robot Plan Execution

**Primary Keywords:** *Multi-Agent Planning*

## Abstract

One area of research in Multi-Agent Path Finding (MAPF) is to determine how re-planning can be efficiently achieved in the case of agents being delayed during execution. One option is to determine a new wait order, i.e., an ordering for multiple agents that are planned to visit the same location, to find the most optimal new solution that can be produced by re-ordering the wait order. We propose to use a Switchable Temporal Plan Graph and a heuristic search algorithm to approach finding a new optimal wait order. We prove the admissibility of our algorithm and experiment with its efficiency in a variety of conditions by measuring re-planning speed in different maps, with varying numbers of agents and randomized scenarios for agents' start and goal locations. Our algorithm shows a fast runtime in all experimental setups.

## Introduction

Multi-Agent Path Finding (MAPF) is the problem of finding a collision-free plan that navigates a team of agents from their start locations to their goal locations. The problem is the core difficulty of numerous practical and industrial applications, such as automated fulfillment and sortation centers (Wurman, D'Andrea, and Mountz 2008; Kou et al. 2019), computer games (Silver 2005; Li et al. 2020), drone swarms (Hönig et al. 2018), and so on. The field of MAPF has garnered considerable interest in recent years. Solving MAPF problems optimally is known to be NP-hard on both graphs and grids (Yu and LaValle 2013; Banfi, Basilico, and Amigoni 2017), and numerous algorithms have been developed to address this challenge.

Classic MAPF models assume flawless execution. However, in real-world scenarios, agents may encounter unexpected delays due to mechanical differences, unforeseen events, localization errors, and so on. To accommodate such delays, existing research suggests the use of a Temporal Plan Graph (TPG) (Hönig et al. 2016). The TPG captures the precedence relationships within a MAPF solution and maintains these relationships during execution. In this model, an agent only advances to the next vertex in its plan if the corresponding precedence conditions are met. Consequently, if an agent experiences a delay, all other agents whose actions depend on the delayed agent will pause. Despite its advantages, the use of TPG can introduce a significant number of waits into the execution results due to the knock-on effect in the precedence relationship.

In this paper, we introduce a variant of the Temporal Plan Graph, named as the Switchable TPG. This model allows for the modification of precedence relationships through two operations on switchable edges, resulting in a new standard TPG. With a Switchable TPG, in the event of a delay, we can generate a new standard TPG that minimizes the cost for all agents to reach their goal locations by executing the new TPG. Subsequently, we propose an optimal heuristic search algorithm to find the new TPG based on a Switchable TPG. We provide proof of the algorithm's optimality and evaluate its efficiency under a variety of experiments. Experimental results show that our approach always finds the optimal TPG with an average runtime faster than 1 second for various numbers of agents on the random-32-32-10 and the warehouse map. On more complicated maps (Paris and Lak303d), our algorithm also runs about 4 times faster than the existing replanning algorithm.

## Problem Definition

We first introduce the formal definition of MAPF.

**Definition 1** (MAPF). Multi-Agent Path Finding (MAPF) is an optimization problem of finding collision-free paths for a team of agents $\mathcal{A}$ on a given graph. Each agent has a unique start location and a unique goal location. Time is discretized into unit-size steps. In each timestep, agents can move to an adjacent location or wait at the current location. A path specifies the actions of an agent at each timestep from its start location to its goal location. We say two agents $i, j \in \mathcal{A}$ collide if either of the following happens:

1. $i$ and $j$ are at the same location at the same timestep.
2. $i$ leaves a location $l$ at a timestep $t$ and $j$ enters the same location $l$ at the timestep $t$.

A MAPF solution is a set of collision-free paths, one for each agent in $\mathcal{A}$.

**Remark 1.** The above definition of collision coincides with that in the setting of *k-robust plan* (Atzmon et al. 2018) with $k$ set to 1. The motivation for disallowing the second type of collision above is that if agents follow each other and the front agent suddenly stops, the following agent may collide with the front agent. Thus this restriction guarantees better robustness when agents are subject to delays.

For our discussion, we will stick to the following format for a MAPF solution, though our algorithm is not dependent on the specific format of the MAPF solution.

**Assumption 1.** A MAPF solution takes the form of a set of paths $\mathcal{P} = \{p_i : i \in \mathcal{A}\}$. Each path $p_i$ is an ordered sequence of location-timestep tuples $(l_0^i, t_0^i) \rightarrow (l_1^i, t_1^i) \rightarrow \cdots \rightarrow (l_{zi}^i, t_{zi}^i)$ with the following properties:

- $t_0^i = 0$. $l_0^i$ is the start location of agent $i$, and $l_{zi}^i$ is its goal location.
- Each tuple $(l_k^i, t_k^i)$ in $p_i$ for $k > 0$ indicates that agent $i$ is planned to perform a move action into the location $l_k^i$ at timestep $t_k^i$. So $t_{zi}^i$ records the time when agent $i$ reaches its goal, i.e. the travel time of $i$.
- We require a temporal ordering of the sequence: $t_k^i < t_s^i$ for all $0 \leq k < s \leq zi$.

These properties force all consecutive pairs of locations $l_k^i$ and $l_{k+1}^i$ to be adjacent on the graph. A wait action is implicitly defined between two consecutive tuples $(l_k^i, t_k^i)$ and $(l_{k+1}^i, t_{k+1}^i)$: if $t_{k+1}^i - t_k^i = \Delta$, then agent $i$ is planned to wait at $l_k^i$ for $\Delta - 1$ timesteps before moving to $l_{k+1}^i$.

We also formalize some phrases related to MAPF: A MAPF solution is **optimal** if it minimizes the sum of travel time for all agents, i.e., $\sum_{i \in \mathcal{A}} t_{zi}^i$. Agents are said to be **executing** a MAPF solution if they act as specified in their paths.

In this paper, we consider the replanning problem when agents are subject to a delay during execution. We model this as Problem 1, parameterized by a delay probability $p$.

**Problem 1.** Given a MAPF solution, at any timestep $t$ during the execution a delay may happen, in which: at least one agent (that hasn't reached its goal) is forced to wait at its current location from timestep $t$ to $t + \Delta$, for some delay length $\Delta$ that may be drawn from some distribution.

When such a delay happens, the delayed agent might block the paths of other undelayed agents and thus hinder their execution. One naïve fix is that once a delay happens, we re-run a MAPF solver with this delay constraint to produce a new solution. However, this approach is usually expensive. Instead, we propose a fast replanning algorithm that lets agents stick to their original location-wise paths but with different move-or-wait sequences.

## Related Works

The field of MAPF has attracted significant attention in recent years. In relation to this, numerous recent studies have explored strategies for managing unexpected delays during execution. A simple strategy to manage unexpected delays is to replan from the beginning. However, this approach is computationally intensive, leading to prolonged agent waiting time. To avoid the need for replanning, Atzmon et al. (2018) suggested the creation of a $k$-robust plan. The approach built robustness into the plan., allowing agents to adhere to their planned paths even if each agent is delayed by up to $k$ timesteps. However, if an agent's delay exceeds

$k$ timesteps, replanning or alternative strategies are still required. Atzmon et al. (2020) then proposed a different approach, computing $p$-robust plans that ensure execution success with a probability of at least $p$, given an agent delay probability model. Nevertheless, planning a $k$-robust or $p$-robust plan is considerably more resource-intensive than computing a standard MAPF plan.

Another strategy for managing delays involves the use of an execution policy to execute a plain MAPF solution, where dependencies or precedence relationships are preserved during execution (Hönig et al. 2016; Ma, Kumar, and Koenig 2017; Hönig et al. 2019). This approach is quick and eliminates the need for replanning. However, the execution results often leave room for improvement, as many unnecessary waits are introduced, and the solution lacks quality guarantees.

## Temporal Plan Graph

Roughly, our algorithm optimizes the *orderings* for multiple agents that are planned to visit the same location. This is achieved using a graph-based abstraction called Temporal Plan Graph.

**Definition 2** (TPG, (Hönig et al. 2016)). A Temporal Plan Graph (TPG) is a directed graph $\mathcal{G} = (\mathcal{V}, \mathcal{E})$ that represents the precedence relationships of a MAPF solution. Given a MAPF solution $\mathcal{P}$, the corresponding TPG is defined as follows:

Each vertex represents a move action of an agent. The set of vertices is $\mathcal{V} = \{v_k^i : i \in \mathcal{A}, k \in [0, zi]\}$, where each vertex $v_k^i$ corresponds to the $k^{\text{th}}$ tuple in the path $p_i$. Each edge encodes a precedence relationship between a pair of move actions. An edge $(u, v)$ encodes that the movement $u$ is planned to happen before the movement $v$. The set of edges $\mathcal{E}$ represents the is partitioned into two types of edges $\mathcal{E}_1$ and $\mathcal{E}_2$.

- **Type 1 edges** connect two vertices corresponding to consecutive tuples for the same agent. The set of Type 1 edges is $\mathcal{E}_1 = \{(v_k^i, v_{k+1}^i) : \forall i \in \mathcal{A}, k \in [0, zi)\}$.
- **Type 2 edges** connect two vertices of distinct agents, as long as they correspond to tuples containing the same location. The set of Type 2 edges is

$$\mathcal{E}_2 = \{(v_{s+1}^j, v_k^i) : \forall i \neq j \in \mathcal{A}, s \in [0, zj), k \in [0, zi]$$
$$\text{satisfying } l_s^j = l_k^i \text{ and } t_s^j < t_k^i\}$$

**Remark 2.** Note that in the above description, we define Type 2 edge as $(v_{s+1}^j, v_k^i)$ instead of $(v_s^j, v_k^i)$ in order to avoid the second type of collision in Definition 1.

**Example 1.** Figure Figure 1 shows an example of converting a MAPF solution into a TPG. Both agents are planned to visit the same location D and the red agent is planned to visit D earlier than the blue agent. Therefore we have a Type 2 edge from $v_3^{\text{red}}$ to $v_2^{\text{blue}}$.

## Executing a TPG

A TPG contains sufficient information for agents' execution. Procedure 1 describes how to execute a TPG in detail, which

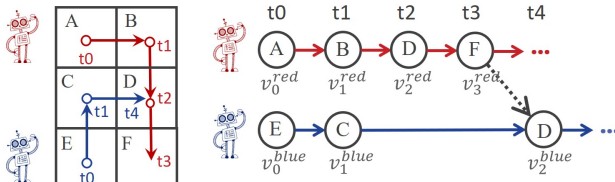

Figure 1: An example of converting a MAPF solution to a TPG. The solid arrows in the TPG represent Type 1 edges, and the dashed arrow represents a Type 2 edge.

includes two helper functions $\text{INIT}_{\text{EXEC}}$ and $\text{STEP}_{\text{EXEC}}$ and a main function EXEC, along with two marks "satisfied" and "unsatisfied" for vertices.

Intuitively, marking a vertex as satisfied corresponds to moving an agent to the corresponding location, and we do so if and only if all in-neighbors of this vertex have already been satisfied. The execution terminates when all vertices are satisfied, i.e. all agents have reached their goals. We now formally state some properties of a TPG. All missing proofs are delayed to the appendix due to the space limit.

**Definition 3** (Deadlock). When executing a TPG as Procedure 1, we say a *deadlock* is encountered iff in an iteration of the while-loop [line 17], the set $\mathcal{S}$ contains only NULL on line 21 yet there exists unsatisfied vertex in $\mathcal{V}$.

**Lemma 1.** *Executing $\mathcal{G}$ encounters a deadlock if and only if there exists a cycle in $\mathcal{G}$.*

Lemma 1 shows a correspondence between a cycle and a deadlock that holds for all TPG. Next, we focus on properties that are specific to a TPG that is constructed from a MAPF solution.

**Proposition 2** (Collision-Free). *Let $\mathcal{G}$ be a TPG constructed from a MAPF solution as in Definition 2. Assuming $\mathcal{G}$ is executed as in Procedure 1 and an agent $i$ is moved to its $k^{th}$ location $l_k^i$ at timestep $t$ iff vertex $v_k^i$ is satisfied in the $t^{th}$ iteration of the while-loop on line 17, any two agents $i, j$ never collide.*

One can also derive the following interesting corollary from the proof of Proposition 2, which is going to be useful later for our algorithm.

**Corollary 3.** *Let $\mathcal{G}$ be a collision-free TPG. If we replace an arbitrary Type 2 edge $(v_{s+1}^j, v_k^i)$ in it with $(v_{k+1}^i, v_s^j)$, the TPG remains to be collision-free.*

Next, we relate the *cost* of a TPG with the sum of travel timesteps of a MAPF solution.

**Proposition 4.** *Let $\mathcal{G}$ be a TPG constructed from a MAPF solution $\mathcal{P}$, the cost of $\mathcal{G}$ is no greater than the sum of travel time for agents following $\mathcal{P}$.*

Proposition 4 gives an immediate corollary that $\mathcal{G}$ is deadlock-free.

**Corollary 5** (Deadlock-Free). *If a TPG $\mathcal{G}$ is constructed from a MAPF solution $\mathcal{P}$, then it is deadlock-free.*

---

**Procedure 1:** Execute $(\mathcal{G} = (\mathcal{V}, \mathcal{E}))$

Lines highlighted in blue are activated to compute the cost of a TPG, and can be omitted for the mere purpose of execution.

1   Define a counter $cost$;
2   **Function** $\text{INIT}_{\text{EXEC}}(\mathcal{G})$
3     $cost \leftarrow 0$;
4     Mark all vertices in $\mathcal{V}_0 = \{v_0^i : i \in \mathcal{A}\}$ as satisfied;
5     Mark all remaining $v \in (\mathcal{V} \setminus \mathcal{V}_0)$ as unsatisfied;

6   **Function** $\text{STEP}_{\text{EXEC}}(\mathcal{G}, i)$
7     **if** $\forall k : v_k^i$ satisfied **then**
8       **return** NULL;
9     $cost \leftarrow cost + 1$;
10    $v \leftarrow v_k^i : v_k^i$ unsatisfied and $\forall k' < k, v_{k'}^i$ satisfied;
11    **forall** $(u, v) \in \mathcal{E}$ **do**
12      **if** $u$ unsatisfied **then**
13        **return** NULL
14    **return** $v$

15   **Function** $\text{EXEC}(\mathcal{G})$
16    $\text{INIT}_{\text{EXEC}}(\mathcal{G})$;
17    **while** there exists unsatisfied vertex in $\mathcal{V}$ **do**
18      Define a set $\mathcal{S} \leftarrow \varnothing$;
19      **forall** agent $i \in \mathcal{A}$ **do**
20        Add $\text{STEP}_{\text{EXEC}}(\mathcal{G}, i)$ into $\mathcal{S}$;
21      **forall** $v \in \mathcal{S}$ **do**
22        **if** $v \neq$ NULL **then**
23          Mark $v$ as satisfied;
24    **return** $cost$;

---

## Switchable TPG

TPG is a handy representation for precedence relationships. However, a standard TPG constructed as in Definition 2 is fixed and bound to a given set of paths. In contrast, our optimization algorithm will use the following extended notion of TPG, which enables flexible modifications of precedence relationships.

**Definition 4** (Switchable TPG). Given a TPG $\mathcal{G}_0$, let $\mathcal{E}_1$ denote its set of Type 1 edges and $\mathcal{E}_2$ denote its set of Type 2 edges, such that $\mathcal{G}_0 = (\mathcal{V}, \mathcal{E}_1, \mathcal{E}_2)$. A switchable variant of $\mathcal{G}_0$ is $\mathcal{G} = (\mathcal{V}, \mathcal{E}_1, (\mathcal{S}_{\mathcal{E}2}, \mathcal{N}_{\mathcal{E}2}))$, which partitions $\mathcal{E}_2$ into two disjoint subsets $\mathcal{S}_{\mathcal{E}2}$ (switchable Type 2 edges) and $\mathcal{N}_{\mathcal{E}2}$ (non-switchable Type 2 edges), and allows two operations with respect to any switchable edge $(v_{s+1}^j, v_k^i) \in \mathcal{S}_{\mathcal{E}2}$:

- $fix(v_{s+1}^j, v_k^i)$ removes this edge from $\mathcal{S}_{\mathcal{E}2}$ and add the same edge into $\mathcal{N}_{\mathcal{E}2}$. i.e., this operation fixes a switchable edge to be non-switchable.
- $reverse(v_{s+1}^j, v_k^i)$ removes this edge from $\mathcal{S}_{\mathcal{E}2}$ and add $(v_{k+1}^i, v_s^j)$ into $\mathcal{N}_{\mathcal{E}2}$. i.e., this operation switches the

precedence relation and then fixes it to be non-switchable.

Figure 2 shows a visualization for the $reverse$ operation.

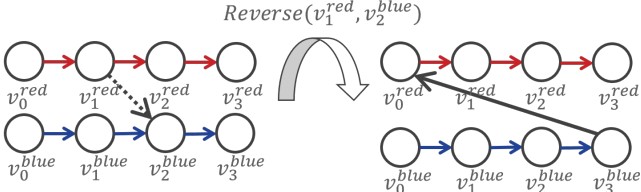

Figure 2: An example of reversing an edge in TPG. After the $reverse$ operation, edge $(v_1^{\text{red}}, v_2^{\text{blue}})$ in the left TPG is replaced with edge $(v_3^{\text{blue}}, v_0^{\text{red}})$ in the right TPG

Definition 4 defines a strict superclass of Definition 2. For clarity, we may refer to a TPG satisfying Definition 2 as a *non-switchable* or a *standard* TPG. A switchable TPG degenerates into a standard TPG if $\mathcal{S}_{\mathcal{E}2}$ is empty, in which case its cost can be determined using Procedure 1. We say a switchable TPG $\mathcal{G}$ *produces* a standard TPG $\mathcal{G}'$ if $\mathcal{G}'$ can be generated through a sequence of $fix$ or $reverse$ operations on $\mathcal{G}$. Note that by Corollary 3, all $\mathcal{G}'$ producible from $\mathcal{G}$ are simultaneously deadlock-free or not.

Then the roadmap of our replanning algorithm is:

(1) when a delay happens, construct a switchable TPG $\mathcal{G}$ corresponding to the current states of agents, and then

(2) produces a standard TPG $\mathcal{G}'$ from $\mathcal{G}$ with the minimum possible cost (among all $\mathcal{G}$-producible $\mathcal{G}'$), such that it represents an optimal ordering of precedence relationships, upon sticking to the original location-wise paths.

We end this section by specifying the construction of a switchable TPG corresponding to a delay situation.

**Construction 1.** Given a MAPF solution $\mathcal{P}$, we execute it by running Procedure 1 on its corresponding standard TPG $\mathcal{G}_0$ as constructed in Definition 2. Assume that at timestep $t$ during the execution, agent $i$ is delayed for $\Delta$ timesteps. We construct a switchable TPG $\mathcal{G}$ for this situation as follows:

1. Let $\mathcal{G} = (\mathcal{V}, \mathcal{E}_1, \mathcal{E}_2)$ be a copy of the standard TPG $\mathcal{G}_0$.
2. Convert it to a switchable TPG$(\mathcal{V}, \mathcal{E}_1, (\mathcal{S}_{\mathcal{E}2}, \mathcal{N}_{\mathcal{E}2}))$ by defining $\mathcal{N}_{\mathcal{E}2} = \{(u,v) : \text{ either } u \text{ or } v \text{ is marked as satisfied in the execution of } \mathcal{G}_0\}$ and $\mathcal{S}_{\mathcal{E}2} = (\mathcal{E}_2 \setminus \mathcal{N}_{\mathcal{E}2})$.
3. Let $v_k^i$ be the next unsatisfied vertex of $i$, i.e. $v_k^i \leftarrow \text{STEP}_{\text{EXEC}}(\mathcal{G}_0, i)$ from Procedure 1. Create $\Delta$-many new dummy vertices: $\mathcal{V}_{\text{new}} = \{v_1, v_2, \cdots, v_\Delta\}$ and $(\Delta+1)$-many new edges:

$$\mathcal{E}_{\text{new}} = \{(v_{k-1}^i, v_1), (v_1, v_2), \cdots, (v_{\Delta-1}, v_\Delta), (v_\Delta, v_k^i)\}.$$

Modify $\mathcal{G}$ such that $\mathcal{V} \leftarrow \mathcal{V} \cup \mathcal{V}_{\text{new}}$ and $\mathcal{E}_1 \leftarrow (\mathcal{E}_1 \cup \mathcal{E}_{\text{new}}) \setminus \{(v_{k-1}^i, v_k^i)\}$.

When executing this TPG, we interpret satisfying a dummy vertex as agent $i$ waiting at location $l_{k-1}^i$ that corresponds to vertex $v_{k-1}^i$.

Construction 1 satisfies the following intuitive yet crucial theorem. The proof is again delayed to the appendix.

**Theorem 6.** *Let $\mathcal{G}$ be a switchable TPG constructed as in Construction 1, there always exists a finite-cost, collision-free standard TPG that can be produced from $\mathcal{G}$.*

## Algorithm

We now describe our algorithm in a top-down modular manner, starting with a high-level heuristic search framework in Algorithm 2. We abuse the operators $fix$ and $reverse$ to take a switchable TPG along with a switchable edge as inputs and return a new TPG after the operation.

---

**Algorithm 2:** Replanning
HEURISTIC, TERMINATE, CYCLEDETECTION, and BRANCH are modules that will be specified later. $\mathcal{X}$ denotes some auxiliary information accompanying a TPG, whose format is defined by the set of modules.

---

**Input:** TPG $\mathcal{G}_{\text{root}} = (\mathcal{V}, \mathcal{E}_1, (\mathcal{S}_{\mathcal{E}2}, \mathcal{N}_{\mathcal{E}2}))$
**Output:** TPG $\mathcal{G}_{\text{result}}$

1 Initialize an empty priority queue $\mathcal{Q}$;
2 $h_{\text{root}} \leftarrow \text{HEURISTIC}(\mathcal{G}_{\text{root}}, \mathcal{X}_{init})$;
3 $\mathcal{Q}.push((\mathcal{G}_{\text{root}}, \mathcal{X}_{init}), 0, h_{\text{root}})$;

4 **while** $\mathcal{Q}$ is not empty **do**
5    $((\mathcal{G}, \mathcal{X}), g, h) \leftarrow \mathcal{Q}.pop()$;
6    $(g', \mathcal{X}', (v_k^i, v_s^j)) \leftarrow \text{BRANCH}(\mathcal{G}, \mathcal{X})$;
7    **if** TERMINATE$(\mathcal{G}, \mathcal{X}')$ **then**
8       $fix$ all edges in $\mathcal{S}_{\mathcal{E}2}$ of $\mathcal{G}$;
9       **return** $\mathcal{G}$;

10    $\mathcal{G}_{\text{f}} \leftarrow fix(\mathcal{G}', (v_k^i, v_s^j))$;
11    **if not** CYCLEDETECTION$(\mathcal{G}_{\text{f}}, (v_k^i, v_s^j))$ **then**
12       $h_f \leftarrow \text{HEURISTIC}(\mathcal{G}_{\text{f}}, \mathcal{X}')$;
13       $\mathcal{Q}.push((\mathcal{G}_{\text{f}}, \mathcal{X}'), g + g', h_f)$;
14    $\mathcal{G}_{\text{r}} \leftarrow reverse(\mathcal{G}', (v_k^i, v_s^j))$;
15    **if not** CYCLEDETECTION$(\mathcal{G}_{\text{r}}, (v_{s+1}^j, v_k^i))$ **then**
16       $h_f \leftarrow \text{HEURISTIC}(\mathcal{G}_{\text{f}}, \mathcal{X}')$;
17       $\mathcal{Q}.push((\mathcal{G}_{\text{r}}, \mathcal{X}'), g + g', h_r)$;

18 **throw exception** "No solution found";

---

Before analyzing Algorithm 2, we recall the notion of cost of a standard TPG and define a similar notion of *partial cost* for a switchable TPG as the cost of its *reduced* standard TPG, which contains only its non-switchable edges.

**Lemma 7.** *Let $\mathcal{G}_{switch}$ be a switchable TPG and $\mathcal{G}$ be an arbitrary standard TPG produced from $\mathcal{G}_{switch}$. The partial cost of $\mathcal{G}_{switch}$ is no greater than the cost of $\mathcal{G}$.*

We prove Lemma 7 in the appendix.

We now state and prove the correctness of our Algorithm 2, under some reasonable assumptions on the modules.

**Assumption 2.** Let $\mathcal{G}_{\text{root}}$ be a switchable TPG constructed as in Construction 1. We assume the modules in Algorithm 2 satisfy the following conditions:

- CYCLEDETECTION$(\mathcal{G}, (u,v))$ returns true iff $\mathcal{G}'$ contains a cycle involving edge $(u,v)$, where $\mathcal{G}'$ is the reduced standard TPG of $\mathcal{G}$ containing only its non-switchable edges.

- Given a TPG $\mathcal{G}$:
  - HEURISTIC($\mathcal{G}, \mathcal{X}$) computes a value $h$.
  - BRANCH($\mathcal{G}, \mathcal{X}$) outputs a value $g'$, an updated auxiliary information $\mathcal{X}'$, and an edge in $\mathcal{S}_{\mathcal{E}2}$ if it is non-empty or NULL otherwise.

  Whenever we push $((\mathcal{G}, \mathcal{X}), g, h)$ into $\mathcal{Q}$, $g + h$ is guaranteed to be the partial cost of $\mathcal{G}$.
- On line 7 of Algorithm 2, TERMINATE returns true iff the partial cost of $\mathcal{G}$ equals the cost of $\mathcal{G}'$, where $\mathcal{G}'$ is produced from $\mathcal{G}$ by $fix$ing all switchable edges.

We make the following observations from Assumption 2:

1. Under the assumption on TERMINATE, when Algorithm 2 reaches line 10, $\mathcal{G}$ must contain switchable edges so $(v_k^i, v_s^j)$ returned by BRANCH is not NULL. This ensures that $fix$ and $reverse$ on line 10 and 14 are well-defined.

2. As long as $\mathcal{G}_{\text{root}}$ is acyclic, it holds inductively that CY-CLEDETECTION($\mathcal{G}, (u, v)$) on line 11 and 15 returns true iff the reduced standard TPG $\mathcal{G}'$ contains *any* cycle. This is because in each iteration, we introduce one new edge into $\mathcal{G}'$, and any new cycle formed in this iteration has to contain this new edge.

**Theorem 8.** *Under Assumption 2, taking $\mathcal{G}_{\text{root}}$ as an input, Algorithm 2 outputs a collision-free standard TPG $\mathcal{G}_{\text{result}}$ with the minimum cost among all possible $\mathcal{G}_{\text{root}}$-producible standard TPG.*

*Proof.* Theorem 6 shows the existence of a solution (in particular, it is collision-free so by Corollary 3, any $\mathcal{G}_{\text{root}}$-producible standard TPG is collision-free.) We now prove the completeness and then admissibility of Algorithm 2.

Algorithm 2 always terminates because by assumption TERMINATE($\mathcal{G}$) returns true once $\mathcal{G}$ contains no switchable edge. There are only finitely many possible operation sequences from $\mathcal{G}_{\text{root}}$ to any standard TPG, each corresponds to an element that can possibly be added to $\mathcal{Q}$. Thus the algorithm must return a solution or report an exception when all possibilities are exhausted after a finite number of steps. We show the completeness using the following claim:

**Claim 9.** *Let $\mathcal{G}'$ be an arbitrary deadlock-free solution that can be produced from $\mathcal{G}_{\text{root}}$. At the beginning of each iteration of the while-loop on line 4, there exists some $\mathcal{G} \in \mathcal{Q}$ such that $\mathcal{G}'$ can be produced from $\mathcal{G}$.*

*Proof (of Claim 9).* This holds inductively: at the beginning of the first iteration, $\mathcal{G}_{\text{root}} \in \mathcal{Q}$. During any iteration, if some $\mathcal{G} \in \mathcal{Q}$ such that $\mathcal{G}$ can produce $\mathcal{G}'$ is popped on line 5, then one of the following must hold:

- $\mathcal{G} = \mathcal{G}'$: $\mathcal{G}$ contains no switchable edge, thus the algorithm terminates in this iteration and the inductive step holds vacuously.
- $\mathcal{G}_f$ produces $\mathcal{G}'$: since $\mathcal{G}'$ is acyclic, so is the reduced TPG of $\mathcal{G}_f$. Thus $\mathcal{G}_f$ won't be pruned by CYCLEDETEC-TION and is added into $\mathcal{Q}$.
- $\mathcal{G}_r$ produces $\mathcal{G}'$: this is symmetric to the previous case.

In any case, the claim remains true after this iteration. $\square$

Claim 9 shows that Algorithm 2 is guaranteed to find a solution if one exists, otherwise the priority queue would remain to be non-empty and the algorithm cannot exit the loop to throw an exception.

Finally, if $\mathcal{G}_{\text{result}}$ is outputted, it must have the minimum cost. Assume towards contradiction that when $\mathcal{G}_{\text{result}}$ is returned, there exists another $\mathcal{G}_0$ in $\mathcal{Q}$ that can produce a standard $\mathcal{G}_{\text{better}}$ with cost smaller than $\mathcal{G}_{\text{result}}$. Yet this is impossible since Lemma 7 implies that such $\mathcal{G}_0$ must have a smaller $g + h$ value and thus would be popped from $\mathcal{Q}$ before $\mathcal{G}_{\text{result}}$. On the other hand, Claim 9 shows that the existence of such a $\mathcal{G}_0$ is the necessary condition for the existence of $\mathcal{G}_{\text{better}}$. Therefore $\mathcal{G}_{\text{result}}$ has the minimum cost.

$\square$

## Execution-based Modules

In this and the subsequent section, we describe two sets of modules and prove that they satisfy Assumption 2. We start with describing a set of "execution-based" modules in Modules 3, which largely ensembles Procedure 1.

**Proposition 10.** *Module 3 satisfies Assumption 2.*

*Proof.* By the property of DFS, CYCLEDETECTION returns true iff there exists a cycle involving edge $(u, v)$. We show by induction that whenever we push $((\mathcal{G}, \mathcal{X}), g, h)$ into $\mathcal{Q}$ in Algorithm 2, $g + h$ equals the partial cost of $\mathcal{G}$.

- In the base case, HEURISTIC computes $h_{\text{root}}$ by running exactly Procedure 1 on the reduced standard TPG of $\mathcal{G}_{\text{root}}$, so $0 + h_{\text{root}}$ is the partial cost of $\mathcal{G}_{\text{root}}$.
- Let $((\mathcal{G}, \mathcal{X}), g, h)$ be popped from $\mathcal{Q}$. Observe inductively that $\mathcal{X}$ records a state of execution of $\mathcal{G}$, and $g$ records the execution $cost$ up to that timestep. On line 6 of Algorithm 2, BRANCH continues execution from that state $\mathcal{X}$ until we reach the next switchable edge (as indicated by the modifications on line 13 of Modules 3). BRANCH outputs the execution cost $g'$ from $\mathcal{X}$ to a current state $\mathcal{X}'$.
  Then line 10 or 14 of Algorithm 2 $fix$ or $reverse$ this edge to get $\mathcal{G}_f$ or $\mathcal{G}_r$, and HEURISTIC on line 12 or 16 of Algorithm 2 continues executing the reduced TPG of $\mathcal{G}_f$ or $\mathcal{G}_r$ from $\mathcal{X}'$ till termination to get $h$. Thus $g + g' + h$ together sums to the total partial cost of $\mathcal{G}_f$ or $\mathcal{G}_r$.

Finally, if on line 7 of Algorithm 2, TERMINATE returns true, then the last BRANCH outputs a goal state for all agents, i.e. the modified execution on line 13 of Modules 3 does not encounter any switchable edge. So it must have been executed on a standard TPG, whose partial cost is exactly its cost after vacuously fixing all its switchable edges. $\square$

## Graph-based Modules

In this section, we introduce an alternative set of modules, which departs from the execution-based ideology of Modules 3 and focus on the graph properties of a TPG. We will see later in our experiment that this shift of focus significantly improves the efficiency of our algorithm. We start by presenting the following crucial theorem that provides a graph-based approach to compute the cost of a TPG.

**Module 3:** Execution-based Modules
---
1 **Function** CYCLEDETECTION($\mathcal{G}, (u,v)$)
2     Run DFS on $\mathcal{G}$ starting from vertex $v$;
3     **if** DFS encounters a cycle **then**
4         $\lfloor$ **return** true;
5     **return** false;

6 Auxillary information $\mathcal{X}$ is a map $\mathcal{X} : \mathcal{A} \to [0 : zi]$, which records the index of the most recently satisfied vertex for each agent;
7 Define $\mathcal{X}_{\text{init}}[i] = 0$ for all $i \in \mathcal{A}$;

8 **Function** BRANCH($\mathcal{G}, \mathcal{X}$)
9     Run Procedure 1 on $\mathcal{G}$ with the following modifications:
   - Change line 4 of INIT$_{\text{EXEC}}$ to:
     Mark all vertices in $\mathcal{V}_0 = \{v_k^i : i \in \mathcal{A}, k \leq \mathcal{X}[i]\}$ as satisfied;
   - Change line 20 of EXEC to:
     **if** $v \leftarrow$ STEP$_{\text{EXEC}}(\mathcal{G}, i) \neq$ NULL **and** $(\exists e \in \mathcal{S}_{\mathcal{E}2} : e := (v, u) \text{ or } (u, v))$ **then**
     **return** $(cost, \mathcal{X}' := \{i \mapsto \text{STEP}_{\text{EXEC}}(\mathcal{G}, i)\}, e)$;
     **else** Add $v$ into $\mathcal{S}$;
   - Change line 24 of EXEC to:
     **return** NULL;

   **return** the output of modified Procedure 1;

10 **Function** TERMINATE($\mathcal{G}, \mathcal{X}$)
11     **if** $\forall i \in \mathcal{A}, \mathcal{X}[i] = zi$ **then**
12         $\lfloor$ **return** true;
13     **return** false;

14 **Function** HEURISTIC($\mathcal{G}, \mathcal{X}$)
15     Define $\mathcal{G}'$ to be a reduced standard TPG containing only non-switchable edges of $\mathcal{G}$;
16     Run Procedure 1 on $\mathcal{G}'$ with the following modification:
   - Change line 4 of INIT$_{\text{EXEC}}$ to:
     Mark all vertices in $\mathcal{V}_0 = \{v_k^i : i \in \mathcal{A}, k \leq \mathcal{X}[i]\}$ as satisfied;
     **return** NULL;

   **return** the output of modified Procedure 1;

---

**Theorem 11.** *Given a TPG, compute the longest path from vertex $v_0^i$ to vertex $v_{zi}^i$ for each $i \in \mathcal{A}$. The sum of lengths of all such longest paths equals the cost of this TPG.*

We again delay the proof for Theorem 11 to the appendix. We adopt the following well-known algorithm to compute the longest paths:

- Compute a topological sort of all vertices in the TPG.
- Given a source vertex $v_0$, assign distance $dist(v_0) = 0$ and $dist(v) = -\infty$ for all $v \neq v_0$.
- For each vertex $v$ in the topological order:

For each vertex $u$ such that $(v, u) \in \mathcal{E}$: if $(dist(u) < dist(v) + 1)$, assign $dist(u) = dist(v) + 1$.

Using this idea, we specify the following set of graph-based modules.

---
**Module 4:** Graph-based Modules
---
1 **Function** CYCLEDETECTION($\mathcal{G}, (u,v)$)
2     $\lfloor$ **return** CYCLEDETECTION($\mathcal{G}, (u,v)$) from Modules 3;

3 Auxillary information $\mathcal{X}$ is a map $\mathcal{X} : \mathcal{V} \to [0, |\mathcal{V}|)$, which records a topological sort of all vertices;
4 Define $\mathcal{X}_{\text{init}}[v] =$ NULL for all $v \in \mathcal{V}$;

5 **Function** BRANCH($\mathcal{G} = (\mathcal{V}, \mathcal{E}_1, (\mathcal{S}_{\mathcal{E}2}, \mathcal{N}_{\mathcal{E}2}))$)
6     Compute $\mathcal{X}'$ to be a topological sort of all vertices $\mathcal{V}$ in the reduced standard TPG of $\mathcal{G}$ (i.e. the sorting consider only the non-switchable edges $\mathcal{E}_1 \cup \mathcal{N}_{\mathcal{E}2}$ in $\mathcal{G}$);
7     **forall** $(u, v) \in \mathcal{S}_{\mathcal{E}2}$ **do**
8         **if** $\mathcal{X}'[u] > \mathcal{X}'[v]$ **then**
9             $\lfloor$ **return** $(0, \mathcal{X}', (u, v))$;
10     **return** $(0, \mathcal{X}', \text{NULL})$;

11 **Function** TERMINATE($\mathcal{G} = (\mathcal{V}, \mathcal{E}_1, (\mathcal{S}_{\mathcal{E}2}, \mathcal{N}_{\mathcal{E}2})), \mathcal{X}$)
12     **forall** $(u, v) \in \mathcal{S}_{\mathcal{E}2}$ **do**
13         **if** $\mathcal{X}[u] > \mathcal{X}[v]$ **then**
14             $\lfloor$ **return** false;
15     **return** true;

16 **Function** HEURISTIC($\mathcal{G}, \mathcal{X}$)
17     Define $\mathcal{G}'$ to be a reduced standard TPG containing only non-switchable edges of $\mathcal{G}$;
18     $h \leftarrow 0$;
19     **forall** agent $i \in \mathcal{A}$ **do**
20         Using the topological sort $\mathcal{X}$ to compute the longest path distance $dist$ from $v_0^i$ to $v_{zi}^i$;
21         $\lfloor$ $h \leftarrow h + dist$ ;
22     **return** $h$;

---

We note that BRANCH and TERMINATE can be easily combined as a single function in the actual implementation. We define them as separate modules for the simplicity of description. We also remark that this set of modules does not use any $g$ values – it completely depends on HEURISTIC to compute the partial cost of the TPG to determine the priority of node exploration.

**Proposition 12.** *Module 4 satisfies Assumption 2.*

*Proof.* By the property of DFS, CYCLEDETECTION returns true iff there exists a cycle involving edge $(u, v)$. It always holds that whenever we push $((\mathcal{G}, \mathcal{X}), g, h)$ into $\mathcal{Q}$ in Algorithm 2, $g + h = 0 + h$ equals the partial cost of $\mathcal{G}$, because HEURISTIC computes the sum of longest paths for all agents, which by Theorem 11 equals the partial cost.

Finally, if TERMINATE($\mathcal{G}, \mathcal{X}$) returns true, then it must be the case that all switchable edges in $\mathcal{G}$ obey the topological sort specified by $\mathcal{X}$. Therefore after fixing all these edges, the cost of the resulting TPG can be computed as the same sum of longest path distance using $\mathcal{X}$, which exactly equals the partial cost of $\mathcal{G}$ □

## Experiment

Proposition 10 and 12 justify using execution-based modules and graph-based modules to implement our algorithm framework. We refer to these two implementations as Execution-based Switchable-Edge Search (ESES) and Graph-based Switchable-Edge Search (GSES). We design a series of experiments to evaluate the efficiency of our algorithms and the quality of their solutions. Our algorithms are implemented in C++ and experiments were performed on a server with a 64-core AMD Ryzen Threadripper 3990X, 192 GB of RAM, and an Nvidia RTX 3090Ti GPU.

We consider the performance of the algorithms on 4 different maps from the MAPF benchmark suite (Stern et al. 2019), with 6 agent group sizes per map. The four maps are random-32-32-20, warehouse-10-20-10-2-1, Paris-1-256, and Lak303d. An visual illustration of them can be found in Figure 4. Regarding each map and group size configuration, we run the algorithms on 25 different scenarios (start/goal locations) with 6 trials per scenario. We set a runtime limit of 90 seconds for each trial. In the experiments, we execute an optimal MAPF solution initially planned by a K-Robust MAPF solver K-Robust CBS (Chen et al. 2021) with $k$ set to 1. At each timestep of the execution, each agent that hasn't reached its goal is subject to a constant probability $p$ of delay. When a delay happens, we draw a delay length $\Delta$ uniformly random from a range $[10, 20]$ and construct a corresponding delay TPG as in Construction 1, and run both replan algorithm ESES and GSES on the TPG. We note that more than one agent may get delayed, in which case Construction 1 naturally generalizes.

### Efficiency

We compare the runtime of our replanning algorithms with replanning using K-Robust CBS. For each map, scenario, number of agents, and delay probability combination, we model 6 random trials for the delay situation and run all three algorithms (ESES, GSES, and K-Robust CBS) on the same situation. Figure 4 plots the resulting runtime. For trials that exceed the 90-second time limit, we count it as 90 seconds in our mean computation. We see that on all maps and delay configurations, GSES runs significantly faster than K-Robust CBS. Most remarkably, on random-32-32-10 and warehouse maps, the runtime of GSES is consistent below 1 second and does not increase significantly when the number of agents increases. This indicates that our algorithm may be of great practical use for real-time replanning applications.

### Comparing ESES and GSES

We observe from Figure 4 that although our two implementations ESES and GSES adopt the same framework, the graph-based approach (GSES) runs significantly faster

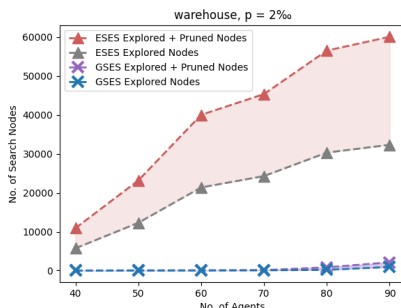

Figure 3: The number of nodes that are explored and pruned by ESES and GSES on the warehouse map. The dashed lines represent the mean value of search nodes. The shaded area between two lines for the same algorithm represents the portion of nodes that are pruned by CYCLEDETECTION.

than the execution-based approach (ESES). This is because the topological sort used in GSES defines a simple but extremely powerful early termination condition (see TERMINATE in Modules 4), which enables GSES to find an optimal solution after a very small number of explorations of the search nodes. We compare the number of search nodes of ESES and GSES on the Paris map with delay probability $p = 0.2\%$ in Figure 3, where *explored node* means the number of TPG that are popped from the priority queue when running our search algorithm, and *pruned node* is the number of TPG that are discarded because CYCLEDETECTION returns true on it.

### Improvement of Solution Cost

We also measure the improvement of our replanned solution, in comparison to the original non-replanned MAPF solution and the replanned solution produced by K-Robust CBS. We stress that our solution is guaranteed to be optimal, as proven in previous sections, upon sticking to the original location-wise paths, while K-Robust CBS finds an optimal solution that is independent of the original paths in the non-replanned solution. Therefore the two algorithms are solving intrinsically different problems, and the results in this section serve primarily for a quantitative understanding of how much improvement we can get by changing only wait orders. Figure 5 plots the mean cost (from the locations where the delay happened to the goal locations) of the replanned-by-GSES, replanned-by-K-CBS, and non-replanned MAPF solutions on four maps with three different delay probabilities. The means are taken across all trials for all different numbers of agents.

We see that the improvements of the solution depend heavily on the map that we operate on. For example, on the random-32-32-10 map, our solutions (which only change the wait orders) have similar costs as the globally optimal solution, while the difference is larger on the Lak303d map.

## Conclusion

This paper proposed a new algorithm framework to find the optimal wait orders for agents that are planned to visit the

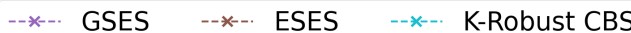

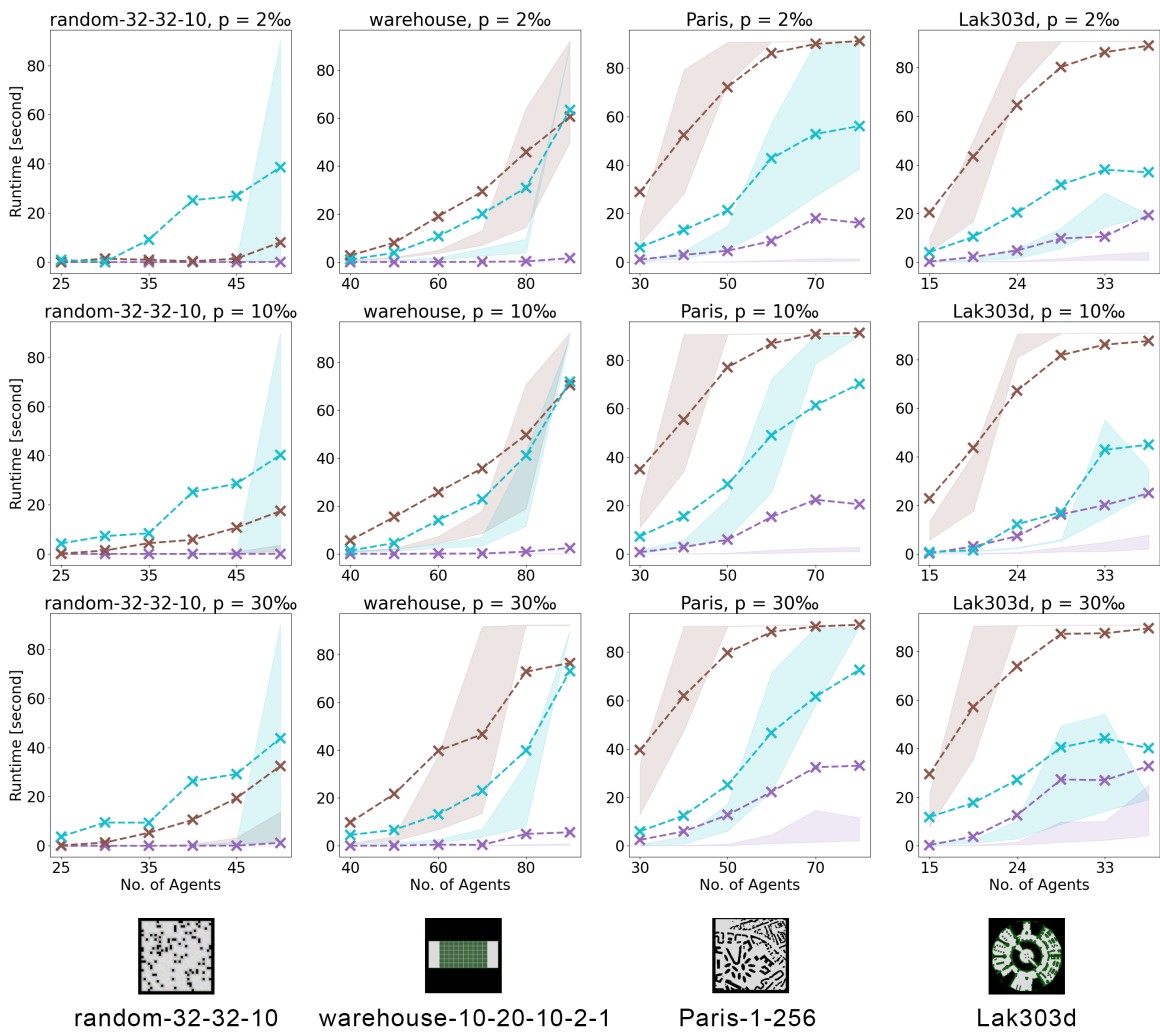

Figure 4: Runtime on random32-32-10, warehouse, Paris, and Lak303d maps measured in second. The dashed lines represent the mean of runtime, and the shaded areas denote the 0.4 to 0.6 quantile range.

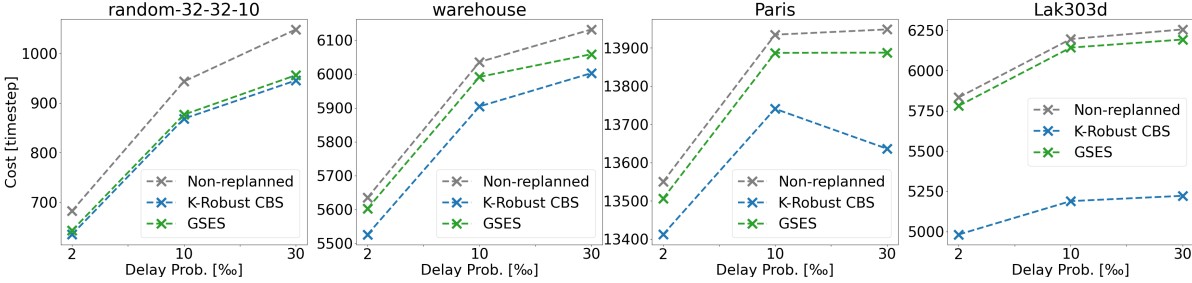

Figure 5: The mean costs of the non-replanned MAPF solution, the ESES-replanned solution, and the K-Robust CBS-replanned solution.

same location. We developed two implementations based on either execution (ESES) or graph presentation (GESE). On the random-32-32-10 and the warehouse maps, the average runtime of GSES is faster than 1 second for various num-

bers of agents. On harder maps (Paris and Lak303d maps), GSES also runs significantly faster than replanning with a K-Robust CBS algorithm.

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
