# OpenReview forum: "A Real-Time Rescheduling Algorithm for Multi-robot Plan Execution"
_icaps-conference.org/ICAPS/2024/Conference — ICAPS 2024_

### Official Review · Reviewer_Fzsy · 2024-01-15

**Significance And Importance:** 2
**Soundness:** 4
**Novelty:** 2
**Clarity:** 3
**Overall Evaluation:** 1
**Confidence:** 3

**Weaknesses:**

0: Minor weaknesses requiring some work to be addressed for the paper to be accepted.

**Contributions Of The Paper:**

The paper presents a replanning algorithm to tackle the Multi Agent Path Finding (MAPF) problem in case of exogenous events occurring during plan execution (i.e., delayed agents).
The main contributions of the paper are the following.
(i) the introduction of a modified version of the concept of Temporal Plan Graph (TPG) called Switchable TPG, built as an extension of the standard TPG upon the acknowledgement of a delay to be represented on the original Graph.
(ii) the introduction of two different implementations (modules) of the procedures necessary to realize the Replanning Algorithm (Algorithm 2), one execution-based, and the other graph-based, respectively called ESES and GSES.
(iii) The introduction of a number of theorems aimed at proving some properties of the modules' functions, necessary to guarantee the applicability of the approach.
(iv) the implementation of a rather complete empirical evaluation, where the performances of the presented replanning procedures are compared against a third competitor approach (K-Robust CBS).

**Ethical Considerations:**

(1) Not Applicable: The paper does not have any ethical considerations to address

**Nomination For Best Paper:**

No

**Questions For Authors:**

1) Algorithm 2: It is not clear how a fix operation on the switchable TPG G' (line 10) may produce a cycle (deadlock) in the reduced TPG obtained from G', unless G' already contained the deadlock. Introducing a deadlock by reversing an edge (line 14) is instead clear. But in general, shouldn't all the TPGs pushed in the queue Q (and hence, all the TPGs popped out of it) be deadlock-free by construction, due to the controls in lines 11 and 15?

2) Page 5, 1st column, observation 2 ("This is because in each iteration, we introduce one new edge into G?"):  What is the new edge, and where is it introduced, exactly? Is it the edge returned by the BRANCH procedure at line 6? In the positive case, wasn't this edge already contained in S_{\epsilon 2}?

**Reproducibility:**

1: Difficult to reproduce because of missing detail.

**Strengths Of The Paper:**

One strenght of the paper is its clarity and readability; the paper's structure is generally clear, though it may be improved, especially when modules 3 and 4 are mentioned in the text without proper introduction, considering also that the related pseudocode is placed on the following page.
Another strength is the fact that the tackled problem is of high academic interest.
Lastly, the proposed solution seems to be rather efficient in terms of replanning speed in presence of several agents, thus making itself suitable for real-time applications.

**Weaknesses Of The Paper:**

The main weakness of the paper is that the introduction of the ESES approach, i.e., the module that uses the Execution Algorithm (Algorithm 1) to implement the replanning algorithm's modules seems rather "artificial". It is truly no surprise that ESES performance is so bad, compared not only to GSES but also to the third competitor, K-Robust CBS (see Figure 4). The approach purely based on the TP Graph's properties is unsurprisingly more efficient than an approach that has to run the whole execution algorithm on the graph multiple times to realize the same behavior as its GSES competitor. To my perspective, the most significant comparison remains the one between GSES and K-Robust CBS, at least regarding runtime. On the other hand, I fully agree with the auhtors that the comparison regarding the solution cost between GSES and K-Robust CBS has little significance, given that the two approaches solve different problems.

MINOR ISSUES:
1) "Execution-based Modules" Section: "Module 3" is used before being introduced. The pseudocode of Module 3 is in the next page, and this leaves the reader a little puzzled.

2) "Execution-based Modules" Section, line 4:"... which largely ensembles Procedure 1". Maybe, Procedure 2 was meant here?

3) In the caption of figure 5, ESES should be changed to GSES.

---

> ### Author Rebuttal · Authors · 2024-01-28
>
> We thank the reviewer for helpful and constructive feedback and will address the editorial comments in the next draft.
>
> > The main weakness of the paper is that the introduction of the ESES approach seems rather "artificial".
>
> We include the execution-based module because, despite its inefficiency compared to the graph-based module, it exhibits multiple unique advantages: Firstly, the execution-based approach motivates our design of the heuristic search framework – once a switchable edge is encountered during the execution, we spawn two search nodes corresponding to the two directions of this edge. It straightforwardly combines the TPG property and the execution procedure. In contrast, the graph-based module is a less intuitive approach that aims at improving the runtime of the algorithm. We will improve our writing to explain the logical progression between the two approaches more explicitly. Secondly and more importantly, the major component of the execution-based module is just simulating the movements of the agents. This simplicity makes the execution-based approach easily adaptable to various MAPF settings such as a continuous-time setting. As long as we have a simulator for execution, we can plug it into our execution-based approach with minimum additional work, and the algorithm would retain the same functionality.
>
> > It is not clear how a fix operation on the switchable TPG G' (line 10) may produce a cycle (deadlock) in the reduced TPG obtained from G'
>
> Our check for deadlock (controls in lines 11 and 15) considers only the non-switchable edges, i.e. edges that have already been fixed or reversed. For G that is popped from the priority queue Q, it is indeed guaranteed that all non-switchable edges in G do not form a cycle. However, this does not necessarily remain true after we add a new non-switchable edge to G (either via fix or reverse). Thus we need to check again on lines 11 and 15 to ensure that the new graph does not contain a non-switchable cycle before pushing it back to Q.
>
> > What is the new edge, and where is it introduced, exactly?
>
> In the statement, G’ is referring to a reduced standard TPG, which contains only its non-switchable edges, i.e. edges in N_{\epsilon2}. It is true that in the positive case, this edge is already contained in S_{\epsilon 2}, but in the observation, we focus primarily on the non-switchable set of edges N_{\epsilon2}.

---

### Official Review · Reviewer_95Zv · 2024-01-22

**Significance And Importance:** 2
**Soundness:** 3
**Novelty:** 2
**Clarity:** 3
**Overall Evaluation:** 1
**Confidence:** 3

**Weaknesses:**

1: Minor weaknesses that are easily fixable.

**Contributions Of The Paper:**

The paper proposes an improved approach to solve the MAPF problem. In particular, it focuses on improving wait orders for the case that two agents want to use the same location at the same time and one needs to wait. They do so by using Switchable Temporal Plan Graphs, a variant of Temporal Plan Graphs, that enable changing the direction of a dependency. Initially, all dependency edges (between two agents) are flexible. During the search they are fixed or reversed (making a successor state depending on the predecessor state of the other agent). This enables some flexibility during the search. The paper proposes two sets of modules that are referenced in the overall procedure, named execution-based and graph-based set of modules.

**Ethical Considerations:**

(1) Not Applicable: The paper does not have any ethical considerations to address

**Nomination For Best Paper:**

No

**Questions For Authors:**

- Would you consider making the paper more self-sufficient my replacing execution-based modules with the actual proofs?

- Have you run this on an actual multi-robot scenario, real robots or at least in a suitable simulation to move beyond purely theoretical results?

**Reproducibility:**

2: Some details are missing, but the paper still appears to be replicable with some effort.

**Strengths Of The Paper:**

The paper presents an interesting novel idea to decompose the MAPF problem and optimize the waiting times.

The paper is written mostly clearly and with a train of thought that a reader can follow. But sometimes it’s still easy to get lost in the various definitions.

**Weaknesses Of The Paper:**

The paper could be improved by focusing on the graph-based modules and then being able to add the proofs directly into the actual paper, instead of having to rely on the additional material. While the comparison between both sets of modules at first sight is convincing, focusing on the comparison to the state of the art is sufficient and more useful. It is clear that work was done for the execution-based modules, but eventually they seem to not help the paper as much.


Minor:
- Add reference to the maps that you evaluated this on, it’s not immediately clear to the reader what, e.g., “the warehouse map” is
- Def 1: “each agent has a unique start location and unique goal location”: that can be misinterpreted for the obvious fact that an agent has a single (thereby unique) start and goal location. Be more specific that it’s about no other agent being at the location at the same time on start and end.
- Def 2: “represents the is…” -> “represents the_n_ is”
- Example 1: Figure Figure
- Module 4: “from Module 3” (not modules 3)
- Figure 4 would benefit from a better explanation

---

> ### Author Rebuttal · Authors · 2024-01-28
>
> We thank the reviewer for helpful and constructive feedback and will address the editorial comments in the next draft.
>
> > Would you consider making the paper more self-sufficient by replacing execution-based modules with the actual proofs?
>
> We include the execution-based module because, despite its inefficiency compared to the graph-based module, it exhibits multiple unique advantages: Firstly, the execution-based approach motivates our design of the heuristic search framework – once a switchable edge is encountered during the execution, we spawn two search nodes corresponding to the two directions of this edge. It straightforwardly combines the TPG property and the execution procedure. In contrast, the graph-based module is a less intuitive approach that aims at improving the runtime of the algorithm. We will improve our writing to explain the logical progression between the two approaches more explicitly. Secondly and more importantly, the major component of the execution-based module is just simulating the movements of the agents. This simplicity makes the execution-based approach easily adaptable to various MAPF settings such as a continuous-time setting. As long as we have a simulator for execution, we can plug it into our execution-based approach with minimum additional work, and the algorithm would retain the same functionality.
>
> We agree that delaying the proof to the appendix makes the paper less self-sufficient. We will improve our writing in the next draft to shorten the text, which hopefully would leave enough space to contain the actual proofs in the main body. If not, we also plan to buy extra pages to include the proofs.
>
> > Have you run this on an actual multi-robot scenario, real robots or at least in a suitable simulation to move beyond purely theoretical results?
>
> In this paper, we focus on proving the optimality of our algorithms and empirically evaluating their runtime performance. We didn’t run the algorithms on an actual multi-robot scenario, because adapting a new algorithm to the real robots involves lots of work that we didn’t manage to complete in this stage of study. However, given the efficiency of our algorithms, it is indeed exciting to see its performance in an actual multi-robot scenario. We think this could be an interesting direction for future work.

---

### Official Review · Reviewer_rhmz · 2024-01-25

**Significance And Importance:** 2
**Soundness:** 4
**Novelty:** 2
**Clarity:** 2
**Confidence:** 3

**Weaknesses:**

0: Minor weaknesses requiring some work to be addressed for the paper to be accepted.

**Contributions Of The Paper:**

The work propose the use of Switchable Temporal Plan Graph and a heuristic search algorithm to approach replanning when finding a new optimal wait order of agents during plan execution in Multi-Agent Path Finding problems/applications.
The proposed system outpeforms existing replanning algorithms.

**Ethical Considerations:**

(1) Not Applicable: The paper does not have any ethical considerations to address

**Nomination For Best Paper:**

No

**Overall Evaluation:**

-1: (weak reject)

**Questions For Authors:**

Intro says "The field of MAPF has garnered considerable interest in recent years." Can you elaborate why is this the case?

TPG reminds me about Petri Nets (PN) and its token simulation, property guarantees (e.g. deadlock, etc). What would you said is the relation of these two aspects and if you have a sense of PN applicability, benefits in this case.

**Reproducibility:**

3: Authors describe the implementation and domains in sufficient detail.

**Strengths Of The Paper:**

Addresses a relevant problem in the P&S community
System provide optimal solutions
Paper provides experimental results in existing benchmark domains

**Weaknesses Of The Paper:**

Presentation: Would benefit from a running example to help communicate the contribution upfront.

The unit-size step assumption in the problem definition is a strong simplification of the problem in general. I understand this might be the traditional scope in the MAPF research, but in several applications (e.g. robotics, game, etc) the temporsl aspect of traversing an environment can't carry that simplification. The scheduling problem becomes much more challenging when travel time are more realistic and not that synced through unit-size steps.  In fact the use of the term 'rescheduling' might not reflect precisely the scope of the paper. The temporal aspect is not as aligned with temporal planning in the scheduling/OR line of research.

I would expect to see a larger variety of planners from the literature being used in the experiment section. Is the  K-Robust CBS the main baseline here, and representative of the state of the art?

The text should be read carefully to improve grammar and punctuation.

---

> ### Author Rebuttal · Authors · 2024-01-28
>
> We use “rescheduling” to stress that our algorithm optimizes an existing schedule by changing the wait orderings. We are also open to suggestions for alternative terms. We will accompany our algorithms with a visualization example in our revision to provide an intuitive explanation.
>
> Recently, researchers have made significant progress in speeding up MAPF algorithms. These algorithms, which were initially limited to handling only dozens of agents, can now manage hundreds of agents. This progress holds great promise for a wide range of applications, attracting more researchers to explore enhanced MAPF algorithms and inspiring practitioners to investigate the generalization of MAPF algorithms to real-world scenarios. Additionally, with recent advancements in robotics, larger multi-robot systems are being deployed in industries, making MAPF a critical challenge that needs to be addressed.
>
> We adopt the discrete-time assumption for simplicity. TPG is compatible with continuous-time settings, for example [Hoenig 2019] introduces additional variables such as vertex status and edge weights to construct a TPG for continuous settings and test it on real robots. However, this adds more complications. Thus we use the simpler model for the clarity of presentation.
>
> Our execution-based module can indeed be applied to the continuous-time setting. Since its major component is simulating the agent’s movement, by adapting the execution simulation to a continuous-time setting, our algorithm should retain the same functionality. One can also imagine applying our graph-based module to a continuous-time setting, yet that might require more work such as encoding the traversal time as appropriate TPG edge weights and computing the heuristics using methods like [Hoenig 2016]. This could be an interesting direction for future work.
>
> When a delay happens during the execution of a MAPF solution, existing work either (1) sticks to the original solution using TPGs (i.e. no replanning) or (2) replans the paths using a MAPF solver. We use “Non-replanned” to represent Method (1) and replanning with K-Robust CBS by [Chen et al 2021], a state-of-the-art optimal MAPF solver, to represent Method (2).
>
> Thanks very much for pointing us to Petri Nets. It seems to resonate with our goal of recording the precedence relationship and ensuring properties such as deadlock-free. We will take a closer look at it.
>
> W. Hoenig, et al. Multi-Agent Path Finding with Kinematic Constraints. ICAPS, 2016.

---

### Meta-Review · Area_Chair_NrzX · 2024-02-04

**Recommendation:** Accept (Oral)
**Confidence:** 4

**Metareview:**

This paper studies the dynamic version of the Multi-Agent Path Finding where the plan needs to be revised in response to delays. It proposes a novel way of reordering the wait orders of multiple agents visiting the same location. The paper introduces a new variant of Temporal Plan Graph, and utilizes it within the proposed optimal heuristic search algorithm. All reviewers agree that the paper is clearly written. In the final version of the paper, the authors should resolve the questions post by the reviewers, including the similarity of their approach with petri nets.

**Ethical Considerations:**

(1) Not Applicable: The paper does not have any ethical considerations to address